# *Globodera pallida* virulence on major potato resistance has a common genetic basis across Western Europe

Arno S. Schaveling[1], Leidy van Rijt[2], Yoonseon Do[1], Nike Soffree[1],
Daan Langendoen[1,3], Hilde Room[3], André Machado Bertran[4], Margien Raven[4],
Sebastiaan P. van Kessel[5], Evelyn Y.J. van Heese[6], Stefan J.S. van de Ruitenbeek[1],
Casper C. van Schaik[1], Sebastian Kiewnick[2], Geert Smant[1], Mark G. Sterken[1]*

**1** Laboratory of Nematology, Wageningen University & Research, Wageningen, the Netherlands,
**2** Julius Kühn-Institut, Institute for Plant Protection in Field Crops and Grassland, Braunschweig,
Germany, **3** Averis Seeds B.V., Valthermond, the Netherlands, **4** Hilbrands Laboratorium B.V., Wijster,
the Netherlands, **5** Dutch General Inspection Service for Agricultural Seeds and Seed Potatoes (NAK),
Emmeloord, The Netherlands, **6** Netherlands Institute for Vectors, Invasive Plants and Plant health
(NIVIP), National Plant Protection Organization (NPPO-NL), NVWA, Wageningen, the Netherlands

\* mark.sterken@wur.nl

ppat.1014201

Riverside, UNITED STATES OF AMERICA

**Peer Review History:** PLOS recognizes the
benefits of transparency in the peer review
process; therefore, we enable the publication
of all of the content of peer review and
author responses alongside final, published
articles. The editorial history of this article is
available here: https://doi.org/10.1371/journal.
ppat.1014201

## Abstract

The potato cyst nematode *Globodera pallida* poses a major threat to potato pro-
duction in Western Europe. Current management strategies largely depend on the
use of potato varieties carrying the genetic resistance $GpaV_{vrn}$. However, reports
from multiple West-European countries indicate a steady rise in virulence against
$GpaV_{vrn}$-containing potato varieties, raising serious concerns about *G. pallida* control.
Although recent studies have resolved the genetic basis of virulence in two Dutch *G.
pallida* populations, it remains unclear how conserved this genetic adaptation is in
populations from different regions. To investigate this, we first selected eight Dutch *G.
pallida* populations on the $GpaV_{vrn}$-containing potato variety Seresta and confirmed
a previously identified virulence locus. Second, by analysing the allele frequencies
of four virulence-associated SNPs in Dutch, British, and French $GpaV_{vrn}$-selected *G.
pallida* populations, we found that the same allele is consistently selected by $GpaV_{vrn}$
across Western Europe. Third, we analysed the propagation of eight *G. pallida*
populations on 26 $GpaV_{vrn}$-containing potato varieties and showed that a population's
allele frequency of a single SNP (T173N) accurately reflects its reproduction on
$GpaV_{vrn}$. Fourth, we developed an allele-specific quantitative PCR (AS-qPCR) assay
to determine a population's alternative allele frequency (AAF) of T173N and showed
that AS-qPCR-based AAFs reliably indicate virulence levels on $GpaV_{vrn}$ in Dutch and
German *G. pallida* populations. Together, these findings suggest that a common
allele is consistently selected by $GpaV_{vrn}$ in populations from different regions across
Western Europe. The AS-qPCR assay developed in this study provides a practical

**Data availability statement:** All scripts and underlying datasets are available through gitlab (https://git.wur.nl/published_papers/schaveling_2026_pallida_asqpcr). The data of presented experiments has been included in supplementary files. The DNA sequencing data of the selection experiments is deposited at BioStudies (E-MTAB-16285).

**Funding:** This work was supported by the PPS subsidies from the Dutch Ministry of Agriculture, Nature and Food Quality and Topsector T&U project PALLIFIT (KV1604-022/TU-16004) to GS, the PPS subsidies from the Dutch Ministry of Agriculture, Nature and Food Quality and Topsector T&U project PALLIGEN (LWV22225/TU202202) to GS and MGS, the German Agency for renewable resources, project "ASPARA" (FKZ 2222NR093B) to SK, HORIZON EUROPE Food, Bioeconomy, Natural Resources, Agriculture and Environment (Grant 1083727; NEM-EMERGE) to AB, SK, GS, and MGS, the Dutch Research Council domain Applied and Engineering Sciences VENI grant (17282) to MGS, and the Dutch Research Council domain Applied and Engineering Sciences VIDI grant (21240) to MGS. The funders had no role in study design, data collection and analysis, decision to publish, or preparation of the manuscript.

**Competing interests:** I have read the journal's policy and the authors of this manuscript have the following competing interests: After the AS-qPCR results were unblinded, one of the private partners involved in the validation, HLB, began considering offering the assay as a service to growers. HLB is the employer of André Machado Bertran and Margien Raven.

tool to estimate *G. pallida* virulence on $GpaV_{vrn}$ in the field, enabling field-tailored and sustainable resistance management strategies for farmers.

## Author summary

Potato cyst nematodes are microscopic worms that live in the soil, infect potato roots and cause major yield losses worldwide. Farmers in Western Europe primarily rely on growing potato varieties that carry resistance genes for their nematode management. To control the potato cyst nematode *Globodera pallida* farmers grow potato varieties that carry $GpaV_{vrn}$. However, some *G. pallida* populations can overcome this resistance, making control less effective over time. To manage and prevent this risk, it is important to understand how these nematodes overcome potato resistances. Here, we show that the same genetic shift takes place in nematode populations from different countries across Western Europe when they overcome $GpaV_{vrn}$-based resistance. We found that the frequency of one genetic variant within a population predicts how well it can reproduce on resistant potato plants. Building on this insight, we developed a DNA-based test (AS-qPCR) that can measure the frequency of this variant in field populations. This tool enables the estimation of nematode virulence, allowing farmers and advisors to make informed field-tailored decisions on *G. pallida* management.

## Introduction

*Globodera pallida* and *G. rostochiensis* are potato cyst nematodes (PCN) that cause world-wide losses in potato cultivation [1]. Since the most effective nematicides have been banned [2], management strategies primarily rely on crop rotation and the use of PCN-resistant potato varieties. Resistances against PCN in cultivated potato are derived from wild *Solanum* species. *Globodera pallida* resistance in commercial potato varieties is primarily based on *GpaV* locus from *Solanum vernei* ($GpaV_{vrn}$; [3,4]). However, selection for virulence by $GpaV_{vrn}$ has led to the rise of resistance-breaking *G. pallida* populations [4–7]).

Potatoes and PCN coevolved in the Andes region [8]. Here, PCN evolved the ability to hatch upon the perception of potato root exudates. After hatching, juveniles search for and penetrate the potato roots and establish a feeding site (syncytium) that supplies them with nutrients. Meanwhile, potatoes developed the ability to detect and react to invading juveniles by activating defence responses, preventing juveniles from reaching maturity [9]. Potato resistances can generally be classified as male-biased (masculinizing) or blocking (female-biased; [10]). A male-biased resistance, such as $GpaV_{vrn}$, acts before the sex determination [11]. It restricts syncytium development, limits a juvenile's nutrient availability, and thereby promotes male-development. Since male-biased resistances allow avirulent individuals to transmit

their alleles to the next generation, this slows down the breakdown of resistance [12,13]. In contrast, a blocking resistance, such as *Gpa2*, may allow initial syncytium formation before triggering localised necrosis in the cells around the syncytium that eventually starves juveniles [14]. As female development is highly nutrient-demanding, starvation mainly affects female development [15].

PCN are thought to be introduced into Europe through the importation of potatoes in the nineteenth century [16–18]. Although all West-European *G. pallida* populations originate from the same region in Southern Peru, they are rich in allelic variation [19]. Since the introduction of $GpaV_{vrn}$-containing potato varieties in the 1990s, these have been extensively used for *G. pallida* control. The deployment of $GpaV_{vrn}$ exerts positive selection pressure favouring nematodes carrying virulence alleles. Over time, the widespread and repeated use of $GpaV_{vrn}$ increased the frequency of these alleles leading to breakdown of resistance [4].

*Globodera pallida* and *G. rostochiensis* are classified as quarantine organisms by the European Commission [20], making them subject to strict phytosanitary regulations. Monitoring of PCN involves both voluntary and mandatory sampling of potato fields [21,22]. Upon the extraction and identification of PCN, DNA can be extracted from cysts to determine the *Globodera* species. In cases of suspected virulence, farmers can have the cysts tested on a panel of PCN-resistant potato varieties to determine which variety to grow. However, these variety choice assays are costly and time consuming. Although, long advocated for [23], no rapid, cost-effective and high-throughput test for the detection and quantification of virulence on $GpaV_{vrn}$ in *G. pallida* has been developed yet.

Recently, a single locus has been associated with virulence on $GpaV_{vrn}$ in two Dutch *G. pallida* populations and a single gene (*Gp-pat-1*) was found to be directly involved [4]. This gene contained four non-synonymous SNPs that significantly correlated with virulence in two *G. pallida* populations. Building on this work, we confirmed the recently identified virulence locus by bulked segregant analysis on eight *G. pallida* populations that were selected on $GpaV_{vrn}$-containing potato variety for four generations. We assessed the allele frequencies of four virulence-associated SNPs in *Gp-pat-1* and found three of these SNPs to be selected across 13 Dutch, French, and British *G. pallida* populations that were all independently selected on potato varieties harbouring $GpaV_{vrn}$. For one of these SNPs the frequency of the alternative allele significantly correlated with reproduction on $GpaV_{vrn}$. With an allele-specific qPCR (AS-qPCR) we were able to accurately quantify the alternative allele frequency (AAF) of this virulence-associated SNP and identify $GpaV_{vrn}$-mediated shifts in the AAF. Finally, AS-qPCR-based AAFs showed to be predictive for virulence on $GpaV_{vrn}$ in Dutch and German *G. pallida* populations.

## Methods

### General notes on data analysis

All analyses were conducted in R (version 4.4.1) using Rstudio (version 1.4.1717; [24]; The R [25]). In R, the *tidyverse* packages, especially *ggplot2* and *dplyr* were used for data processing in general and generation of figures [26]. Other, specific packages used are mentioned at the relevant sections. All R scripts and underlying data are available through git (https://git.wur.nl/published_papers/schaveling_2026_pallida_asqpcr). Sequencing data of the 9 selected and unselected *G. pallida* AMPOP populations was deposited at the European Nucleotide Archive (E-MTAB-16285).

### Selection of *G. pallida* populations on potato variety Seresta and re-sequencing

We generated new sequencing data for nine *G. pallida* populations that were obtained from Dutch potato fields from 2011 until 2015, including the previously described AMPOP10 and eight other populations: AMPOP03, -06, -08, -09, -10, -13, -15, -16, and -19. After initial propagation on Desiree, these populations were selected on the $GpaV_{vrn}$-containing potato variety Seresta, as previously described [4]. In short, populations were propagated either for four generations on the resistant potato variety Seresta (S4), or, as a control, for one generation on the susceptible variety Desiree (D; S1 Table). Populations resulting from these selection steps were taken as input for DNA isolation and sequencing.

First, DNA extraction and sequencing was performed as previously described [4]. In short, for each S4 and D population DNA of approximately 40 cysts was extracted with phenol-chloroform extraction. DNA was sequenced by BGI Genomics at approximately 200x coverage using DNBSeq with 150 bp paired-end reads [4]. For sample AMPOP06D, 27.6 million duplicated reads were removed from FASTQ files with SeqKit (v2.10.0; [27]), after which all data was uploaded to ENA (E-MTAB-16285). This data was supplemented with two pairs of read data for AMPOP02D versus AMPOP02S4 and one pair of AMPOP10D versus AMPOP10S4. These datasets were included in the analysis and previously published by Schaveling et al. [4]. The four independently generated sequencing libraries of AMPOP02 and AMPOP10 were used for method validation.

## Bulk segregant analysis

For analysis of genetic variation the reads were mapped to the Rookmaker genome (PRJEB91928; [4]). Variant calling was performed using Bcftools (mpileup & call, v1.14; [28]) as previously described ([4]; https://git.wur.nl/stefan.vanderuitenbeek/dnaseq_variant_calling_snakemake_pipeline). Before bulk segregant analysis, the resulting variant calls were filtered by removing sites with QUAL < 50, variants with a mean read depth outside two standard deviations of the sample average or a minimum read depth of 30, insertions, deletions, and sites with low allelic variation (mean alternative allele frequency ≤ 0.05 or ≥ 0.95). Bulk segregant analysis was performed by conducting a chi-squared test on the read depths for the reference and alternative alleles in the selected and non-selected populations. This is similar to the G-statistic [29], however as we sequence deep and aim for a high minimum coverage, we calculate the statistic depending on the actual read depths. The resulting p-values were FDR-corrected using the p.adjust function, as we are including variants with linkage disequilibrium.

To identify loci under selection, we counted the number of significant variants in consecutive windows of 10 kb across each scaffold. Identification of bins with a larger number of associated variants was done based on the significance of the numbers based on an exponential distribution. To fit the distribution well, the number of counts per bin was square-root transformed. Bins exceeding the significance threshold (p-value < 0.05) were defined as putative virulence loci.

## Sequencing data of Dutch, British and French populations

Previously, we identified four non-synonymous SNPs inside the coding sequence of a gene involved in virulence on $GpaV_{vrn}$ (S2 Table; [4]). To assess how well the AAF of these four SNPs correlate with selection on $GpaV_{vrn}$, we used four different DNA sequencing data sets. First, we used the sequencing data of a previous selection experiment with two *G. pallida* populations (AMPOP02 and AMPOP10) selected on the $GpaV_{vrn}$-containing Seresta (E-MTAB-15408; [4]). Second, we used the sequencing data of the nine $GpaV_{vrn}$-selected *G. pallida* populations described above (E-MTAB-16285). Third, we obtained DNA sequencing data of two French $GpaV_{vrn}$-selected populations (Saint-Malo and Noirmoutier) from the ENA (PRJEB90550; [30]). Both populations were selected on the $GpaV_{vrn}$-containing potato variety Iledher, the $GpaV_{spl}$-containing genotype 96D31.51, and the susceptible variety Desiree, *in duplo* for ten consecutive generations. Fourth, we obtained DNA sequencing data of a British *G. pallida* population Newton from the ENA (PRJEB41175; [31]). The Newton population was selected for 12 generations on the $GpaV_{vrn}$-containing genotype Sv_11305 (Morag) and the *H3*-containing CPC2802 (Sa_11415) genotype. Mapping and variant calling on the *G. pallida* Rookmaker genome was performed as described above.

## Correlating SNP data with phenotypic data

Two standard PCN resistance tests were conducted in accordance with the [32]. In short, two litre pots with a potato plant were inoculated with approximately 10,000 living larvae. For each population the exact number was determined. At the end of the experiment (after 3 months the plants were watered for the last time and thereafter dried for ~ 1 month), cyst numbers were counted per pot. Per pot, the average number of larvae per cyst was determined and multiplied by the total

number of cysts in the pot to get the final number of larvae per pot. The reproduction rate was calculated by dividing the final number of larvae ($P_f$) by the number of inoculated larvae ($P_i$):

$$Reproduction\ rate = \frac{P_f}{P_i}$$

A total of 8 *G. pallida* populations were tested across two tests (S1 Table). Both tests included AMPOP02, -03, -06, -09, -10 and -16. In addition, test 1 also included AMPOP13 and -15. Test 1 included fourteen potato varieties, that were grouped into clusters based on their resistance level (as determined previously by [4]). All varieties carry *GpaV$_{vrn}$*. Six varieties were included from the Seresta cluster (CI$_{SER}$; Axion, Ardeche, Arsenal, VD 07–0289, Innovator, and Seresta) and eight of the Festien cluster (CI$_{FES}$; Avarna, Avito, Altus, Supporter, Basin Russet, Libero, HZD 06–1249, and Festien). Test 2 included eight varieties from CI$_{SER}$ (Actaro, Aveka, Novano, Avatar, Simphony, Stratos, Vermont, and Seresta) and six from CI$_{FES}$ (BMC, Merenco, Saprodi, Sarion, Sereno, and Festien). Each tests contained at least three biological replicates per population (S3 Table). The average reproduction on Seresta of both tests was used as an indication of the virulence level on *GpaV$_{vrn}$*.

### An AS-qPCR for the quantification of the alternative allele frequency of T173N

To quantify the AAF of SNP T173N (AAF$_{T173N}$) in *G. pallida* field samples, we decided to develop an allele-specific (AS) qPCR assay. Therefore, we first aimed at specifically amplifying the reference and the alternative alleles. We developed two sets of primers: one standard set and one with a locked nucleic acid (LNA) at the 3' end. Both allele-specific primers had the SNP at the 3' end (S4 Table). We calculated the qPCR-based AAFs based on the amplification of the alternative allele relative to the total amplification [33] according to the formula:

$$AAF = \frac{2^{-Cq\_ALT}}{2^{-Cq\_ALT} + 2^{-Cq\_REF}} = \frac{1}{1 + 2^{(Cq\_ALT - Cq\_REF)}}$$

with *Cq_REF* being the Cq-value of the primer pair amplifying the reference allele and *Cq_ALT* being the Cq-value of the primer pair amplifying the alternative allele, reducing the need for housekeeping genes.

To obtain DNA for the AS-qPCR, 60–70 cysts were crushed with bead-beating in 150 µL water for 80 seconds at 30 Hz. Lysis was performed with an *in-house* lysis buffer [34]. DNA was extracted using the Maxwell RSC PureFood GMO and Authentication Kit in combination with the Maxwell RSC instrument (Promega, USA) following the manufacturer's instructions. This yielded between 4.1 and 11.3 ng/ µL of DNA per sample. AS-qPCRs were performed using iQ Supermix (Bio-Rad, USA; S5 Table) on a CFX96 C100 Touch thermal cycler (Bio-Rad, USA; S6 Table).

In a first test, these primers were tested on seven *G. pallida* populations, including three *in house* reference populations (D383, Rookmaker and AMPOP10) and four populations that were recently isolated from a field with suspected breakdown of *GpaV$_{vrn}$* resistance. As the standard allele-specific primers gave a better indication of the AAF than the LNA primers, we continued our analysis with the standard primers. In a second test, the standard primers were tested on an additional six recently isolated *G. pallida* field populations.

### Blinded test of the AS-qPCR across four laboratories

To validate the AS-qPCR, we designed a blinded test. This test was conducted across four labs: Dutch General Inspection Service for Agricultural Seeds and Seed Potatoes (NAK), Hilbrands Laboratorium (HLB), Nederlandse Voedsel- en Warenautoriteit (NVWA), Wageningen University and Research (WUR). Each lab tested the same blinded set of 12 *G. pallida* populations (S1 Table). This included 10 *GpaV$_{vrn}$*-unselected and selected populations and two reference populations (D383 and Rookmaker). Each of the labs used its own DNA extraction protocol and qPCR machines. DNA extraction was performed on 20–40 cysts.

The WUR performed the AS-qPCR as described above, but with DNA isolation on 20 cysts, yielding between 101.7 and 281.8 ng/µL of DNA per sample. The NAK crushed the cysts with bead-beating in 100 µL water for 5 minutes at 30 Hz. Lysis was done by adding 300 µL Tissue & Cell Lysis Solution (LGC BioSearch Technologies, USA) and 1 µL proteinase K for 15 minutes at 65 °C shaking at 750 rpm. After 10 minutes on ice, 150 µL pre-cooled MPC Protein Precipitation Solution (LGC BioSearch Technologies, USA) was added and centrifuged at 16,000 rcf for 10 minutes. DNA and RNA was isolated with magnetic beads in a KingFisher Flex system (ThermoFisher Scientific, USA), yielding between 5.7 and 38.7 ng/µL of DNA per sample. AS-qPCRs were performed with PerfeCTa SYBR Green SuperMix (QuantaBio, USA) on an Applied Biosystems 7500 Real-Time PCR system (ThermoFisher Scientific, USA). The HLB crushed the cysts, after which lysis and DNA purification was carried out using the Nexttec Tissue & Cell DNA Isolation Kit (Nexttec, Germany) according to the manufacturer's instructions. This yielded concentrations ranging from 8.5 to 97.5 ng/µL. AS-qPCRs were performed with SYBR Green PCR Mix (Clear Detections, Netherlands) on a CFX96 thermal cycler (Bio-Rad, USA). The NVWA crushed the cysts with bead-beating in a mix of 50 µL water, 130 µL ATL buffer (Qiagen, Germany) and 20 µL proteinase K for 2 minutes at 30 Hz. Lysis was done at 56 °C for 60 minutes shaking at 800 rpm. DNA extraction was performed with the DNeasy Blood and Tissue Kit (Qiagen, Germany) according to the manufacturers protocol, yielding DNA concentration ranging between 2.4 and 13.8 ng/µL. AS-qPCRs were performed with GoTaq qPCR Master Mix (Promega, USA) on a CFX Opus Real-Time PCR machine (Bio-Rad, USA).

The methods on the assessment of the reliability of the AS-qPCR assay can be found in the S1 Text.

### Using Dutch and German *G. pallida* populations to assess the predictive value of the AS-qPCR assay

To assess the predictive power of the AS-qPCR, six Dutch *G. pallida* field populations were tested on five potato varieties, including the susceptible variety Desiree and four *GpaV*$_{vrn}$-containing varieties (Seresta, Festien, Axion, and Supporter) in small container tests as previously described [4]. Cysts were counted at 8–12 weeks after inoculation. Relative susceptibilities were calculated for each resistant variety by dividing the number of cysts on a resistant variety by the number of cysts on a Desiree times 100%.

To assess the value of the AS-qPCR on German *G. pallida* populations, virulence levels of ten German *G. pallida* populations obtained from the Emsland region (Germany) were determined through bioassays routinely used [35]. In short, eye plugs of the susceptible potato variety Desiree or the *GpaV*$_{vrn}$-containing varieties Seresta and Axion were individually planted and inoculated with cysts. White females were counted at six to eight weeks after inoculation. The relative susceptibility (RS) was calculated by dividing the number of white females on resistant varieties by the number of white females on the susceptible control times 100%. Using this type of bioassay allows for fast evaluation of the virulence of a *G. pallida* population [36], however, assays that are more artificial can give aberrant estimates of virulence compared standardized pot tests [37]. However, previously we saw good correlations between closed container tests and pot tests for *G. pallida* virulence on *GpaV*$_{vrn}$ [4]. All ten German *G. pallida* populations were tested by the AS-qPCR assay. From each population, DNA was extracted from 60 cysts using the MasterPure Complete DNA & RNA Purification Kit (Lucigen, USA) following the manufacturers protocol. The DNA was suspended in 10mM Tris-HCl buffer and DNA concentration was determined by a Qubit 4 fluorometer (Life Technologies, Singapore). AS-qPCRs were performed with SSoAdvanced SYBR Green Supermix using a CFX 96 Real-Time PCR machine (Bio-Rad, USA), with 14–23ng template DNA per 20µl reaction in triplicate.

## Results

### Bulk segregant analysis on eight *G. pallida* field populations independently verifies a virulence locus

Previously, we identified a locus that strongly associates with virulence in two *G. pallida* field populations [4]. This locus lies on scaffold 28 of the *G. pallida* Rookmaker genome, with the strongest association between 6,371,140 – 6,682,119 bases. For the identification we conducted whole genome sequencing on each generation of a five-generation selection experiment on two *G. pallida* populations (AMPOP02 and AMPOP10) with one to three replicates per population per

generation. The locus was identified based on association of allele-frequency shifts over these five generations. Here we aimed to independently verify that this locus is involved in virulence on $GpaV_{vrn}$. To make that feasible, we confirmed in four independent pairs of selected AMPOP02 and AMPOP10 whether bulk-segregant analysis (BSA) on the sequencing pools with only an unselected and a four-generation selected population as input was sufficient to identify the locus in all four tests (S1 Fig). From this experiment, we conclude that BSA based on generations of selection is a feasible approach for locus verification.

To independently verify the selected locus, we propagated eight *G. pallida* field populations on the $GpaV_{vrn}$-containing variety Seresta for four generations and sequenced the genome pool of starting and final populations (Fig 1A). This included *G. pallida* populations AMPOP03, -06, -08, -09, -13, -15, -16, and -19. Next, we performed bulk segregant analyses (BSAs) on each of the eight *G. pallida* populations separately. We identified the previous associated locus in six of the eight populations (S2 Fig). To be able to draw an overall conclusion from these separate analyses, we pooled the individual outcomes (number of significantly associated variants per 10kb bin). Over all eight populations we identified a single locus on scaffold 28 from 6.21 – 7.80 Mb (p < 0.05; Fig 1B). The 95% strongest associated loci (>477 significant SNPs counted over the 8 selected populations per 10kb bin) were located between 6.46 – 6.69 Mb. Since the previously

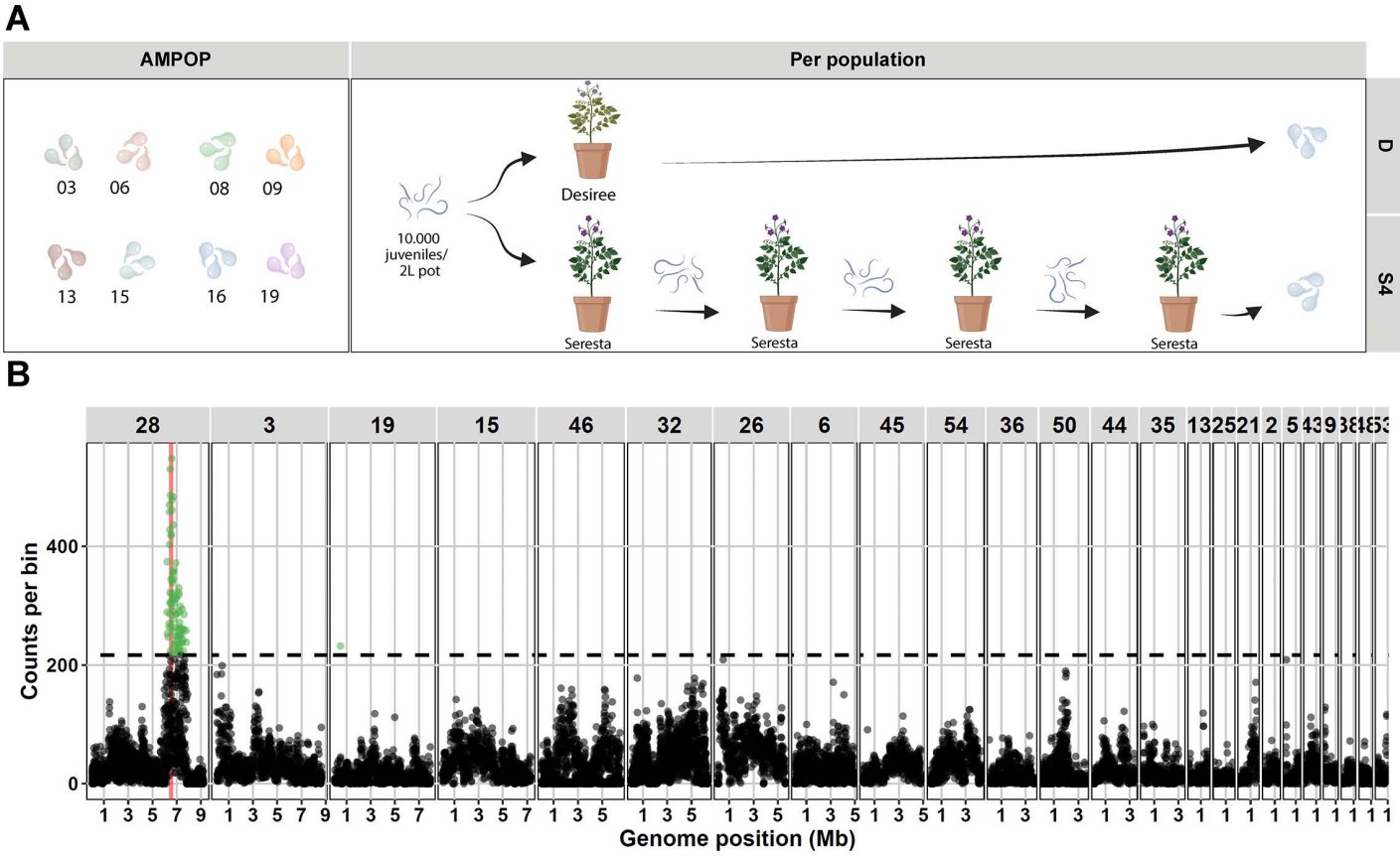

**Fig 1. Bulked segregant analysis confirms the previously identified virulence locus. A** A graphical representation of the eight *G. pallida* AMPOP populations and the selection experiment. Created in BioRender (BioRender.com/v1ycgcu). **B** The number of significant SNPs (FDR < 0.05) are plotted per 10kb bin over eight bulk segregant analyses. Bins that are significantly enriched (p < 0.05) are coloured in green. The virulence locus is indicated in red.

identified virulence locus on scaffold 28 ranged from 6.37-6.68 Mb [4], we conclude this analysis independently verifies the virulence locus.

## Three SNPs are consistently selected by $GpaV_{vrn}$ across Dutch, British and French *G. pallida* populations

Following verification of the virulence locus, we aimed to investigate whether *G. pallida* virulence on $GpaV_{vrn}$ has a common genetic basis. Previously, we identified a single gene on the virulence locus, *Gp-pat-1*, as a gene that contributes to the breakdown of $GpaV_{vrn}$ [4]. Four non-synonymous SNPs (E98V, T173N, L562V, L562S) inside the coding sequence of *Gp-pat-1* correlated with virulence on $GpaV_{vrn}$ in two *G. pallida* populations (S2 Table; [4]). To identify allelic variants involved in or tightly linked to virulence on $GpaV_{vrn}$, we assessed the alternative allele frequencies (AAFs) of these four SNPs in unselected and $GpaV_{vrn}$-selected *G. pallida* populations. First, we assessed the AAFs in two previously selected *G. pallida* populations (AMPOP02 and 10; [4]). Over five generations of selection, the four SNPs showed a consistent average increase in AAF of 0.041 per generation (Fig 2A). Second, we compared the AAFs of these four SNPs in nine *G. pallida* populations (AMPOP03, -06, -08, -09, -10, -13, -15, -16, and -19). These populations were selected on the $GpaV_{vrn}$-containing potato variety Seresta for four generations and compared with the unselected populations. For T173N, L562V, and L562S we observed a consistent and significant increase in the AAFs (p-value ≤ 0.00022; Fig 2B). This indicates that we have now observed a similar increase in the AAF for three SNPs in a total of 10 genetically unique Dutch *G. pallida* populations.

Under similar experimental conditions, French and British *G. pallida* populations have shown the potential to break $GpaV_{vrn}$ resistance as well [38,39]. Two French populations (Saint-Malo and Noirmoutier) were selected on the $GpaV_{vrn}$-containing potato variety Iledher for ten generations [30]. A British population (Newton) was selected on $GpaV_{vrn}$-containing genotype Sv_11305 (Morag) for twelve generations [31]. To determine the AAFs of the four SNPs in these unselected and $GpaV_{vrn}$-selected *G. pallida* populations, we mapped their sequencing data (PRJEB90550 and PRJEB41175) to the *G. pallida* Rookmaker genome. Like the Dutch populations, the French and the British populations showed a significant increase in the AAF for the same three SNPs (Fig 2C). In conclusion, three SNPs (T173N, L562V, and L562S) are consistently selected by $GpaV_{vrn}$ across 13 Dutch, British and French *G. pallida* populations.

## The alternative allele frequency of a single SNP correlates with the reproduction rate on $GpaV_{vrn}$

The consistent correlation of three SNPs with selection on $GpaV_{vrn}$ indicated that these SNPs are either tightly linked to or causal for virulence. To test whether any of our four SNPs gives a robust prediction of virulence, we assessed the reproduction rate (Pf/Pi) of eight *G. pallida* field populations (AMPOP02, -03, -08, -09, -10, -13, -15, and -16) on 28 potato varieties across two standard PCN resistance tests [32]. These potato varieties were previously grouped into three clusters based on their level of *G. pallida* resistance. These clusters were named after the susceptible potato variety Desiree ($Cl_{DES}$), and the $GpaV_{vrn}$-containing potato varieties Seresta ($Cl_{SER}$) and Festien ($Cl_{FES}$; [4]). The PCN tests included two $Cl_{DES}$ varieties, thirteen $Cl_{SER}$ varieties and thirteen $Cl_{FES}$ varieties.

Since virulence on $GpaV_{vrn}$ is a recessive trait [40], only individuals homozygous for the alternative allele (*aa*) are assumed to be virulent, whereas genotypes *Aa* and *AA* will be avirulent. If we assume that the populations are in Hardy-Weinberg equilibrium, the proportion of *aa* genotypes is expected to equal the square of the AAF. Therefore, to assess a potential quadratic relationship, we plotted the $AAF^2$ against the average reproduction rates on $Cl_{DES}$, $Cl_{SER}$, and $Cl_{FES}$. No significant correlations were found between the average reproduction rate on $Cl_{DES}$ varieties and the $AAF^2$ of the four SNPs (Fig 3). This indicates that the AAFs are not predictive for the reproduction rate on susceptible varieties, which is in line with our expectations. For $Cl_{SER}$, the AAF of T173N significantly correlated with the average reproduction rate on thirteen varieties (p = 0.004). Moreover, when analysed individually, of the six $Cl_{SER}$ potato varieties that were tested by eight *G. pallida* populations, five showed a significant correlation between the AAF of T173N and the reproduction rate (p < 0.05; S3A Fig). For the other three SNPs, no significant correlations with reproduction on $Cl_{SER}$ were found. For $Cl_{FES}$, the

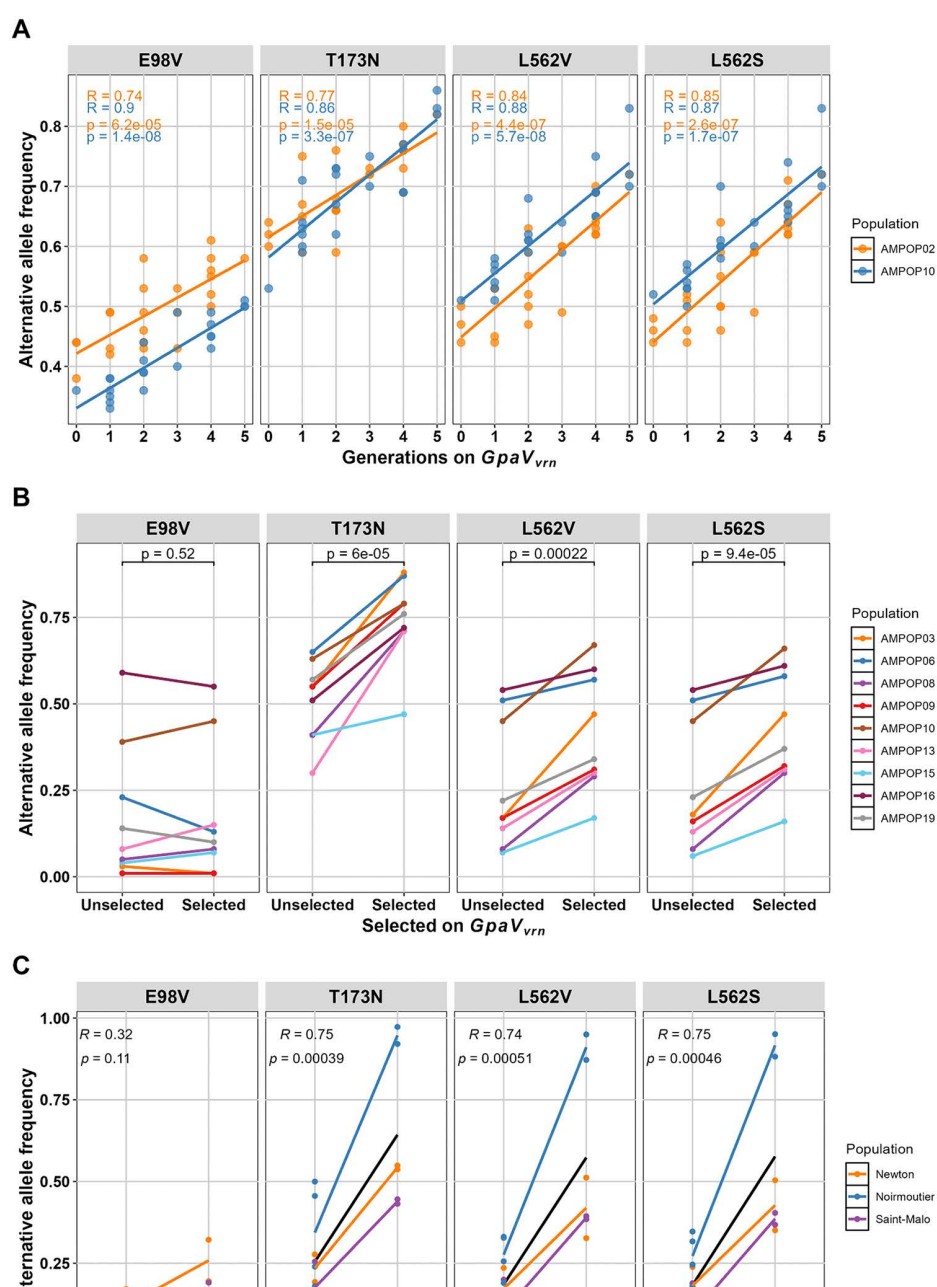

**Fig 2. The alternative allele frequencies of four SNPs that are located within the coding sequence of *Gp-pat-1*. A** The alternative allele frequencies (AAFs) of two *G. pallida* populations selected for five generations on the *GpaV$_{vrn}$*-resistant Seresta potato variety. AAFs show a significant increase over time. Do note that the y-axis is not starting at zero. **B** Nine *G. pallida* field populations were selected for four consecutive generations. SNPs T173N, L562V, and L562S show a significant correlation with selection. Statistics are based on one-sided paired t-tests. **C** Two French *G. pallida* populations (Noirmoutier and Saint-Malo) and an English *G. pallida* population (Newton) show a similar increase in the AAF over time when selected on a *GpaV$_{vrn}$*-resistant potato variety for 10 and 12 generations, respectively. Black lines indicate linear regression models. Statistics in **A** and **C** are based on linear regression models.

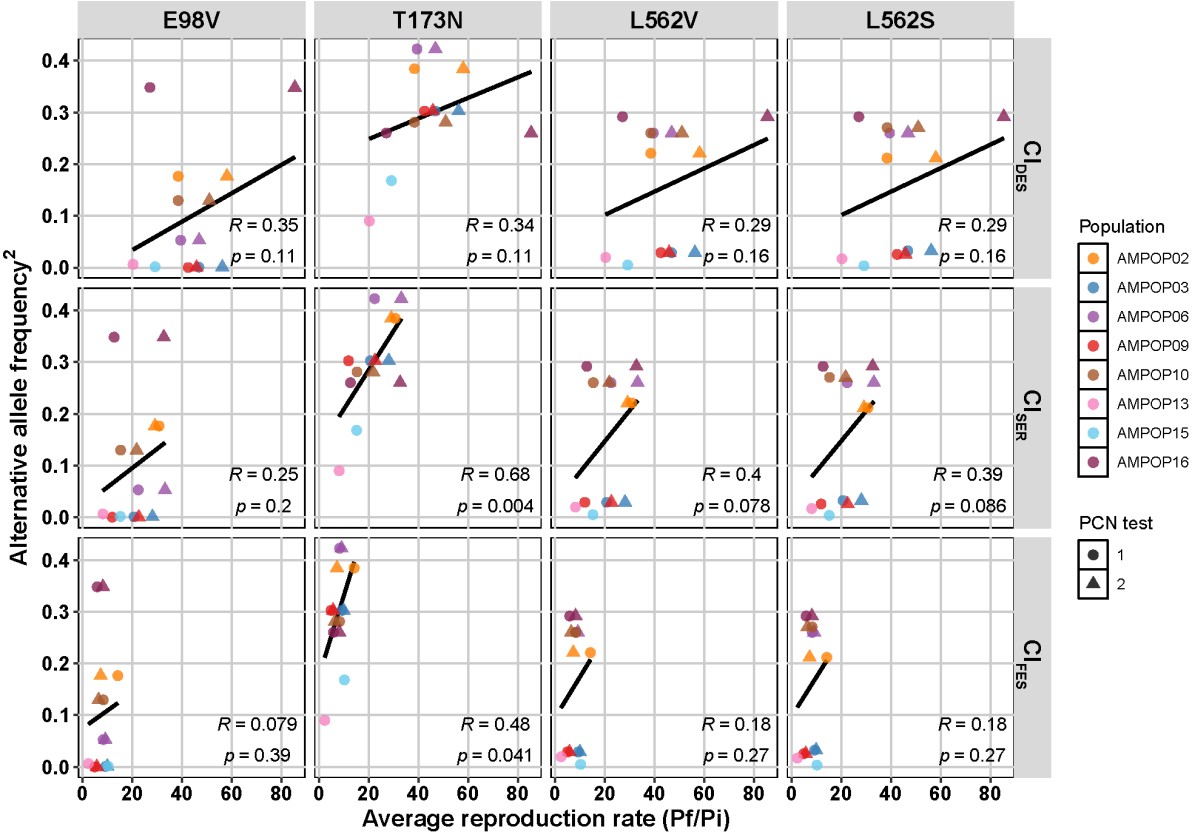

**Fig 3. A population's alternative allele frequency of T173N correlates well with its average reproduction rate on *GpaV_vrn*.** The allele frequencies of eight Dutch *G. pallida* field populations were plotted against their average reproduction rates (Pf/Pi) on two susceptible varieties from the Desiree cluster (CI_DES), thirteen *GpaV_vrn*-containing potato varieties from the Seresta cluster (CI_SER) and thirteen *GpaV_vrn*-containing potato varieties from the Festien cluster (CL_FES). Significant correlations between the AAF of T173N and the average reproduction on CI_SER and CI_FES indicate that a population's allele frequency of T173N is a good prediction for its virulence level on *GpaV_vrn*. Statistics are based on linear regression models that are indicated by black lines.

average reproduction rate also showed a significant correlation with the AAF of T173N (p = 0.041; Fig 3). However, when analysed individually, only in two of the eight CI_FES varieties tested with eight *G. pallida* populations showed a significant correlation between the AAF of T173N and the reproduction rate (p < 0.05; S3B Fig). Together, this data does show that a population's AAF for T173N gives an accurate prediction of its average virulence level on *GpaV_vrn*-containing CI_SER and CI_FES varieties, making a promising quantitative genetic marker for virulence.

## Allele-specific qPCR confirms *GpaV_vrn*-mediated selection of T173N

We established that the alternative allele of T173N is consistently selected by *GpaV_vrn* and the allele frequency in a population correlates with a population's virulence level on *GpaV_vrn*. Subsequenly, we aimed at developing a PCR-based assay to quantify the AAF of T173N in *G. pallida* field populations. We therefore designed allele-specific primers to specifically amplify the avirulent reference allele and the virulent alternative allele in a qPCR (S4a Fig and S4 Table). We set out to verify the reliability and applicability of this assay (S1 Text).

First, we tested two variants of the same allele-specific primer sequences: one with a 3'-terminal locked nucleic acid (LNA), and one standard primer. The AAFs as determined by both primer pairs (AAF_qPCR) significantly correlated with sequencing-based AAF estimates (AAF_seq; p ≤ 0.0014; S4b Fig). However, the LNA-modified primer overestimated the

PLOS Pathogens

AAFs, particularly at lower AAFs. In contrast, the standard primers provided an accurate reflection of the sequencing-based AAFs across the tested range.

Second, we validated the standard AS-qPCR primers on thirteen *G. pallida* populations, including three previously described populations (D383, Rookmaker and AMPOP10) and ten field populations that we recently isolated from Dutch potato fields. Linear regression models confirmed a strong correlation between the $AAF_{qPCR}$ and the $AAF_{seq}$ (p = 1.5e-10; Fig 4A) and the regression coefficients ($\beta_1$ = 1.00 and $\beta_0$ = 0.031) indicated that the AS-qPCR assay reliably estimates AAFs across the full AAF range.

Third, we assessed the robustness of this test. Across four labs we tested the same set of 22 *G. pallida* populations, including the unselected and Seresta-selected AMPOP02 and AMPOP10 populations [4], our eight unselected and Seresta-selected AMPOP populations, and reference populations D383 and Rookmaker. The resulting $AAF_{qPCR}$ values of each lab showed highly significant correlations with the sequencing data and with the $AAF_{qPCR}$ values of the other three labs (Fig 4B). This indicated that the assay is robust and can accurately quantify the AAF. When we compared the $AAF_{qPCR}$ of the unselected populations with the $GpaV_{vrn}$-selected populations, nine out of ten $GpaV_{vrn}$-selected populations showed a significant increase in their AAF of T173N as compared to the corresponding unselected population (p < 0.05; Fig 4C). Fourth, we tested the AS-qPCR on six *G. pallida* populations previously propagated on potato roots in small container tests. A significant correlation between a population's average relative susceptibility on $GpaV_{vrn}$-containing potato varieties and its $AAF^2$ (R = 0.83, p = 0.021) confirmed that the AS-qPCR is predictive for virulence on $GpaV_{vrn}$ (S5A Fig). Fifth, we tested the AS-qPCR on ten German *G. pallida* populations. These populations were propagated on the potato varieties Desiree, Seresta, and Axion. A significant correlation between a populations $AAF^2$ and the relative susceptibility on Seresta (R = 0.61, p = 0.031) indicates that the AAF of SNP T173N makes a quantitative genetic marker for virulence on $GpaV_{vrn}$ in German *G. pallida* populations (S5B Fig). Together, this confirms that $GpaV_{vrn}$-mediated selection acts on T173N in each of the tested populations and the AAF of T173N represents a population's stage of selection.

## Discussion

Here, we identified a virulence allele in *G. pallida* that is consistently selected across 13 West-European *G. pallida* populations. We showed that its allele frequency (AAF) correlates with a population's reproduction on $GpaV_{vrn}$, and developed an allele-specific qPCR (AS-qPCR) assay predictive for virulence levels of *G. pallida* field populations.

### The allele frequency of a single SNP is a predictor for a population's level of virulence on $GpaV_{vrn}$

Through bulked segregant analyses (BSA) on eight $GpaV_{vrn}$-selected and unselected *G. pallida* populations we identified a major virulence locus on scaffold 28 of the Rookmaker genome (Fig 1B). This confirms the findings of two previous studies in which *G. pallida* virulence on $GpaV_{vrn}$ was associated with the same locus on scaffold 28 and the syntenic region on scaffold 2 of the D383 genome [4,30]. Collectively, three independent studies, each using a distinct statistical approach (BSA, pairwise $F_{ST}$, association analyses) on separate pools of $GpaV_{vrn}$-selected *G. pallida* populations, converged on the same virulence locus. In addition to the locus on scaffold 28, we found other loci to be under selection. As these scaffolds are less consistently selected, they may have population-specific contributions to virulence. Given that this is the third independent study to identify the same locus on scaffold 28, strongly supports its central role in virulence on $GpaV_{vrn}$.

We then assessed the alternative allele frequencies (AAFs) of four SNPs from *Gp-pat-1*, a gene residing in the associated locus and previously associated with virulence on $GpaV_{vrn}$ in two *G. pallida* populations [4]. We found that three of the four SNPs were consistently selected by $GpaV_{vrn}$ across eight other Dutch, one British, and two French *G. pallida* populations (Fig 2). This indicates that these SNPs in *Gp-pat-1* are either causal for or tightly linked to virulence on $GpaV_{vrn}$. If these SNPs are linked to causal variants, the strength of the correlation between the AAF and virulence will depend on the degree of recombination between the SNP and the causal allelic variant. Across two PCN resistance tests with eight *G. pallida* field populations, we showed that the AAF of SNP T173N significantly correlates with the average reproduction

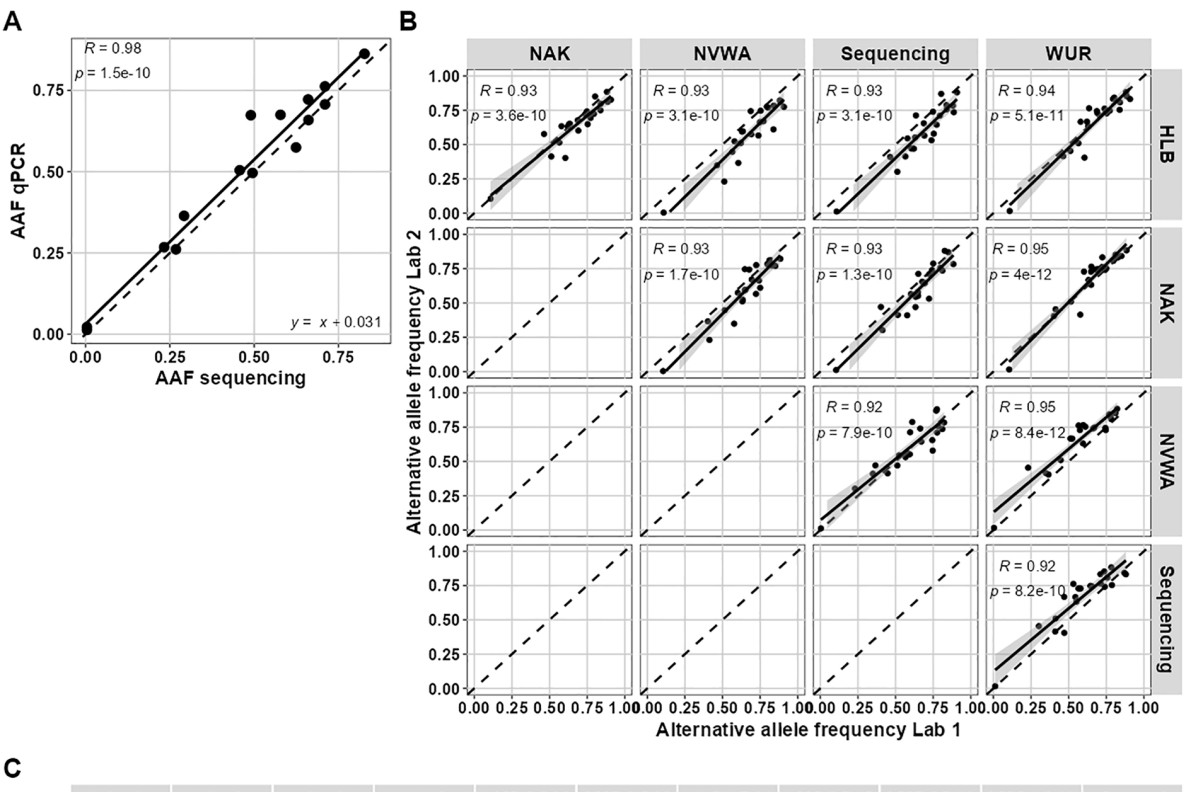

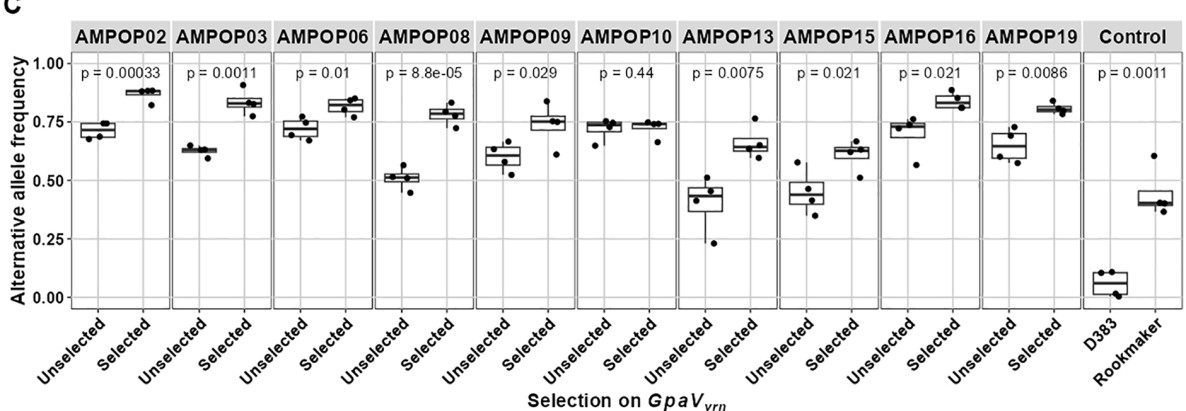

**Fig 4. An allele-specific qPCR identifies *GpaV_vrn*-mediated increases in the alternative allele frequency. A** The alternative allele frequency (AAF) of SNP T173N in 13 *G. pallida* populations as measured with allele-specific (AS) primers in a qPCR against the AAF according to DNA sequencing. **B** The validation of the AS-qPCR assay at four different labs. AAFs as determined by AS-qPCR by one lab are plotted against the AAFs as determined by qPCR by the other lab or against the AAF as determined by sequencing. Statistics in **A** and **B** are determined by linear regression models. Black dashed lines indicates y = x for visual support. **C** The AAFs of unselected and selected *G. pallida* populations as determined by qPCR in four different labs. Unselected populations were propagated on the susceptible variety Desiree and selected populations were propagated for four rounds on the *GpaV_vrn*-containing variety Seresta. Statistics are based on one-sided t-tests.

rate on thirteen Seresta-like and thirteen Festien-like potato varieties (Fig 3). However, when assessing the correlations for single potato varieties, part of them is insignificant (S2 Fig). This likely reflects additional resistances in $CI_{FES}$ varieties and technical variation inherent to the PCN resistance assay [32]. This is supported by the observation that the only $CI_{SER}$ variety tested twice (Seresta), showed the strongest correlation (p = 0.00057). Given the technical variation associated

with PCN resistance tests, the AAF of T173N may provide a better indication of a population's average virulence level than individual PCN resistance tests. Therefore, the AAF of T173N represented a reliable indicator of *G. pallida* virulence levels in field populations.

To predict virulence in *G. pallida* field populations, we developed an allele-specific qPCR assay that accurately quantifies the AAF of the T173N SNP and distinguishes populations based on their stage of selection (Fig 4). The developed qPCR assay was based on standard primers, where a test with LNA primers, unexpectedly, gave less consistent results (S4 Fig). It is possible that the LNA primers can outperform the standard primers as much lower DNA concentrations than tested [41]. Namely, this tool was developed and validated with samples of 20 cysts per DNA isolation. This was done with the potential for integration in current practices in mind. The tool has therefore not been tested for sensitivity beyond these constraints (e.g., single cysts or single juveniles). Within the scopes of the tested use, the AAF of *G. pallida* field populations was predictive for their propagation on $GpaV_{vrn}$-containing potato varieties and the AS-qPCR was able to accurately determine the AAFs. Therefore, the AS-qPCR assay can serve as a useful tool to predict virulence on $GpaV_{vrn}$ in field populations.

## Translating allele frequencies to field-tailored advice for potato farmers

With an accurate molecular tool for determining the virulence level of a *G. pallida* population, the major challenge lies in the translation of AS-qPCR data to actionable field predictions. For simplicity, we assumed that our populations are in Hardy-Weinberg equilibrium. We observed significant linear correlations between the $AAF^2$ in field populations and their reproduction on $GpaV_{vrn}$ (Figs 3, S6, and S5A) supporting a quadratic relation between a population's AAF of T173N and its virulence levels on $GpaV_{vrn}$. Since the unselected populations were propagated on Desiree before genotyping the assumption that our populations are in Hardy-Weinberg equilibrium is valid for the unselected populations. However, our field populations have been under selection of $GpaV_{vrn}$, which has masculinizing activity [11], distorting the equilibrium. When assuming perfect masculinizing resistance, from the second generation onwards, the AAF is expected to show a linear relationship with the number of females and the reproduction rate from the second generation onwards [40]. However, in the field, perfect masculinizing resistance is not realistic. The ratios in which the allele combinations are present in a population depend on many factors, including the escape rate, the male fraction among virulent individuals and the total fraction of juveniles that is able to reach maturity [12,40,42]. The relationship between the AAF and the reproduction rate is therefore expected to be in between a linear and a quadratic relationship. Moreover, the effect of the AAF on the reproduction could depend on the infection density and the allele frequency. For example, at low AAFs, having the alternative allele may confer a strong selective advantage, while at higher AAFs intraspecies competition might play a more important role, reducing the benefit of having the virulence allele. Since the overall impact of an infestation depends on both the virulence level of the cysts and their density in the field, translating AS-qPCR-based AAF estimates into actionable field predictions requires integrating AS-qPCR data with cyst densities and phenotypic field data. Future work should therefore establish how infection densities and virulence levels translate into crop loss under field conditions.

Accurate AS-qPCR-based predictions of *G. pallida* virulence levels in the field can be of great help for potato farmers. For example, besides the mandatory PCN sampling [21], the majority of Dutch starch potato growers have their fields voluntarily sampled and tested for PCN [22]. Upon the detection of *G. pallida*, farmers can request variety selection assays to determine which variety to grow based on the *G. pallida* in their fields. However, these assays are costly and time-consuming. Given that soil samples with suspected PCN infestation are already (q)PCR-tested for species and viability determination [43], the infrastructure is there to incorporate a qPCR for virulence determination. In combination with existing qPCR tests, our AS-qPCR test may support service providers in offering farmers informed advice on the sustainable deployment of resistances in the field, tailored to field-specific conditions in a cost-effective manner.

**The same virulence allele is selected by _GpaV_vrn across Western Europe**

Our analyses showed that the same allele is consistently selected across 13 Dutch, British and French _G. pallida_ populations. Moreover, the allele frequency of this same allele correlated with reproduction on _GpaV_vrn in Dutch and German _G. pallida_ populations. This strongly supports our hypothesis that _G. pallida_ virulence on _GpaV_vrn has a common genetic basis across Western Europe. As the genetic population structure across Western Europe is homogenous, this can also be the case for the spread of the virulence allele. The spread of virulence alleles suggests that the emergence of _GpaV_vrn resistance-breaking populations can also be expected in West-European countries where no resistance-breaking populations have yet been found. Moreover, since Europe has functioned as a secondary distribution centre for the invasion of PCN into all other continents [16], the spread of virulence alleles is probably not geographically restricted. This implies that the repeated deployment of _GpaV_vrn consistently selects the same virulence locus in _G. pallida_, leading to the rise of similar resistance-breaking _G. pallida_ populations irrespective of geography.

## Conclusion

Here, we showed that (1) the same allele is consistently selected by _GpaV_vrn in _G. pallida_ populations across Western Europe, (2) a population's AAF of SNP T173N is a good prediction of its reproduction rate on _GpaV_vrn, (3) a population's AAF of T173N can be accurately determined by AS-qPCR, and (4) the AS-qPCR can distinguish different stages of selection. Together, this indicates that virulence on _GpaV_vrn has a common genetic basis across Western Europe We conclude that the AS-qPCR-based AAF of SNP T173N accurately reflects a population's relative virulence level on _GpaV_vrn, which makes it the first qPCR-based method for predicting virulence in a parasitic nematode. This assay enables field-specific guidance for sustainable resistance deployment, prolonging the agronomical lifetime of potato varieties, and allows mapping of virulence distribution at different scales. It therefore offers practical value for farmers, service providers, breeding companies, and policymakers.

## Supporting information

**S1 Text. An assessment of the reliability and applicability of the AS-qPCR assay.**
(PDF)

**S1 Table. An overview of the _Globodera pallida_ populations used in this study, indicating which sequence files correspond to which population, the variety the population was propagated on, the total number of reads generated (in millions), the total number of reads that mapped (in millions), the median coverage on the Rookmaker genome, the comments, and which populations were used for which experiment (Selection experiment, BSA test set, Independent BSA, PCN resistance test 1 and 2, AS-qPCR).**
(XLSX)

**S2 Table. An overview of the four SNPs located inside the coding sequence of _Gp-pat-1_ that were previously associated with virulence on _GpaV_vrn.** With the following columns: SNP (name of the single nucleotide polymorphism); Pos. on Rookmaker scaff. 2 (the position of the SNP on the _G. pallida_ Rookmaker genome); ref (the nucleotide on the reference genome); alt (the nucleotide on the alternative allele); aa_pos (the position of the amino acid change caused by the SNP); ref_aa (the amino acid encoded by the reference allele); alt_aa (the amino acid encoded by the alternative allele); p_fdr_AMPOP02 (the FDR-corrected p-value of the SNP as identified in the association analysis on AMPOP02); p_fdr_AMPOP10 (the FDR-corrected p-value of the SNP as identified in the association analysis on AMPOP10); annotation (the annotation of the SNP); Pos. on D383 scaff. 2 (the position of the SNP on the _G. pallida_ D383 genome).
(XLSX)

**S3 Table. Data of the two standardized PCN resistance tests.** A total of eight *G. pallida* populations were tested on 28 potato varieties. With the following columns: PCN test (the number of the PCN test); Population (the name of the *G. pallida* population tested); Variety (the potato variety on which the population is tested); Repeat (the number of the replicate); Pi_average (the average number of pre-parasitic juveniles (ppJ2s) inoculated per pot); cysts_(the number of cysts counted at the end of the experiment); ppJ2 (the total number of pre-parasitic juveniles (ppJ2s) in the cysts that were harvested at the end of the experiment); Pf/Pi (the reproduction rate determined by dividing the total number of ppJ2s at the end of the experiment by the number of ppJ2s inoculated); cluster (the cluster that the potato variety belongs to, as previously determined in [4]).
(XLSX)

**S4 Table. A list of the oligonucleotides used in the AS-qPCR.**
(XLSX)

**S5 Table. The reaction setup of the AS-qPCR.**
(XLSX)

**S6 Table. The details of the AS-qPCR program.**
(XLSX)

**S1 Fig. Pairwise bulk segregant analyses between two unselected and *GpaV$_{vrn}$*-selected populations.** The number of significant SNPs (FDR < 0.05) are plotted per 10kb bin over eight bulk segregant analyses. Bins that are significantly enriched (p < 0.05) are coloured in green. The virulence locus is indicated in red. This was analysed for **A** and **B** AMPOP02, **C** and **D** AMPOP10.
(TIF)

**S2 Fig. Pairwise bulk segregant analyses between eight unselected and *GpaV$_{vrn}$*-selected populations.** The number of significant SNPs (FDR < 0.05) are plotted per 10kb bin over eight bulk segregant analyses. Bins that are significantly enriched (p < 0.05) are coloured in green. The virulence locus is indicated in red. This was analysed for: **A** AMPOP03, **B** AMPOP06, **C** AMPOP08, **D** AMPOP09, **E** AMPOP13, **F** AMPOP15, **G** AMPOP16, and **H** AMPOP19.
(TIF)

**S3 Fig. The alternative allele frequencies (AAF) of eight Dutch *G. pallida* field populations were plotted against their corresponding reproduction rates (Pf/Pi) on A six different *GpaV$_{vrn}$*-containing potato varieties from the Seresta cluster and B eight *GpaV$_{vrn}$*-containing varieties from the Festien cluster.** The AAF of T173N significantly correlates with virulence on four of the six Cl$_{SER}$ varieties and 2 of the 8 Cl$_{FES}$ varieties, indicating that the allele frequency of T173N is a good indication for virulence on *GpaV$_{vrn}$*. Statistics are based on linear regression models that are indicated by black lines.
(TIF)

**S4 Fig. A An overview of the allele-specific (AS) qPCR design.** Template DNA is depicted in blue, primers are depicted in magenta, and DNA amplification in green. Created in BioRender (https://BioRender.com/09y6291). **B** Comparison of two sets of allele-specific qPCR primers: one using LNA-modified primers, with a 3'-terminal locked nucleotide, and one using standard primers. While alternative allele frequency (AAF) estimates from LNA primers significantly correlate with sequencing-based AAFs, they tend to overestimate the AAF in a population, particularly at lower AAFs. In contrast, standard qPCR primers provide a more accurate reflection of the sequencing-based AAF across the tested range. Dashed lines indicate y = x and are added for visual purposes. Statistics are based on linear regression models.
(TIF)

**S5 Fig. (A) The allele frequencies of six Dutch *Globodera pallida* field populations were plotted against their median relative susceptibility (%) on four $GpaV_{vrn}$-containing potato varieties.** Virulence levels were tested in a small container test and taken relative to the susceptible potato variety Desiree. (**B**) The allele frequencies of ten German *G. pallida* field populations were plotted against their mean relative susceptibility (%) on the $GpaV_{vrn}$-containing potato varieties Seresta and Axion. Virulence levels are based on counts of white females in small container tests and taken relative to the susceptible potato variety Desiree. The significant correlations between the AAFs of SNP T173N and the average relative susceptibilities indicates that a population's allele frequency of T173N is a good prediction for its virulence level on $GpaV_{vrn}$. Statistics are based on a linear regression model that is indicated by the black line.
(TIF)

**S6 Fig. The observed quadratic relation between the standard error and the mean alternative allele frequency (AAF) indicates that the reliability of the assay depends on the AAF of the population (Solid black line).** The dashed grey lines indicate the expected standard errors when using halve (n = 10) or double (n = 40) as many cysts in the AS-qPCR.
(TIF)

## Acknowledgments

The authors want to thank the private partners in the PALLIFIT and PALLIGEN project for their support and their constructive attitude towards this project. We want to thank Paul Heeres for his role in selecting the AMPOPs on Seresta. We want to thank Kostas Gaitanis for the first attempt to design allele specific qPCR primers. We would like to thank Hemanth Konigopal for technical support in generating high quality template DNA from German *G. pallida* populations. We want to thank three growers for making their fields available to us to collect cysts to use in the validation. We want to thank the Nem-Emerge consortium for discussions regarding the paper.

## Author contributions

**Conceptualization:** Arno S. Schaveling, Geert Smant, Mark G. Sterken.

**Data curation:** Stefan J.S. van de Ruitenbeek.

**Formal analysis:** Arno S. Schaveling, Mark G. Sterken.

**Funding acquisition:** Geert Smant, Mark G. Sterken.

**Investigation:** Arno S. Schaveling, Leidy van Rijt, Yoonseon Do, Nike Soffree, Daan Langendoen, Hilde Room, André Machado Bertran, Margien Raven, Sebastiaan P. van Kessel, Evelyn Y.J. van Heese, Stefan J.S. van de Ruitenbeek, Casper C. van Schaik, Sebastian Kiewnick, Mark G. Sterken.

**Project administration:** Geert Smant, Mark G. Sterken.

**Resources:** Leidy van Rijt, Daan Langendoen, Hilde Room, André Machado Bertran, Margien Raven, Sebastiaan P. van Kessel, Evelyn Y.J. van Heese, Sebastian Kiewnick.

**Software:** Arno S. Schaveling, Stefan J.S. van de Ruitenbeek, Mark G. Sterken.

**Supervision:** Geert Smant, Mark G. Sterken.

**Visualization:** Arno S. Schaveling, Mark G. Sterken.

**Writing – original draft:** Arno S. Schaveling, Geert Smant, Mark G. Sterken.

**Writing – review & editing:** Arno S. Schaveling, Leidy van Rijt, Yoonseon Do, Nike Soffree, Daan Langendoen, Hilde Room, André Machado Bertran, Margien Raven, Sebastiaan P. van Kessel, Evelyn Y.J. van Heese, Stefan J.S. van de Ruitenbeek, Casper C. van Schaik, Sebastian Kiewnick, Geert Smant, Mark G. Sterken.

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
