## [Decision Letter · Decision Letter 0]

15 Feb 2026

PPATHOGENS-D-26-00016

*Globodera pallida* virulence on major potato resistance has a common genetic basis across Western Europe

PLOS Pathogens

Dear Dr. Sterken,

Thank you for submitting your manuscript to PLOS Pathogens. After careful consideration, we feel that it has merit but does not fully meet PLOS Pathogens's publication criteria as it currently stands. Therefore, we invite you to submit a revised version of the manuscript that addresses the points raised during the review process.

We look forward to receiving your revised manuscript.

Kind regards,

Adler R. Dillman, Ph.D.

Academic Editor

PLOS Pathogens

Shou-Wei Ding

Section Editor

PLOS Pathogens

Sumita Bhaduri-McIntosh

Editor-in-Chief

PLOS Pathogens

orcid.org/0000-0003-2946-9497

Michael Malim

Editor-in-Chief

PLOS Pathogens

orcid.org/0000-0002-7699-2064

**Additional Editor Comments:**

We are pleased to inform you that your manuscript has been favorably reviewed. All three reviewers commended the study for its rigor, clarity, and the significant contribution the developed AS-qPCR assay makes toward nematode management and understanding virulence evolution. Before final acceptance, there are several minor revisions required to strengthen the manuscript. Please pay particular attention to the reviewers' requests regarding the justification of statistical assumptions (specifically the Hardy-Weinberg equilibrium), the clarification of control population handling, and the discussion regarding LNA primer sensitivity. Additionally, please consider moderating the geographic claims in the text to precisely reflect the sampled regions, and incorporate the suggested structural changes to the Results and Discussion sections to enhance readability.

**Journal Requirements:**

At this stage, the following Authors/Authors require contributions: Arno S. Schaveling, Leidy van Rijt, Yoonseon Do, Nike Soffree, Daan Langendoen, Hilde Room, André Machado Bertran, Margien Raven, Sebastiaan P. van Kessel, Evelyn Y.J. van Heese, Stefan J.S. van de Ruitenbeek, Casper C. van Schaik, Sebastian Kiewnick, Geert Smant, and Mark G. Sterken. Please ensure that the full contributions of each author are acknowledged in the "Add/Edit/Remove Authors" section of our submission form.

https://journals.plos.org/plospathogens/s/submission-guidelines#loc-parts-of-a-submission

4) We do not publish any copyright or trademark symbols that usually accompany proprietary names, eg ©,  ®, or TM  (e.g. next to drug or reagent names). Therefore please remove all instances of trademark/copyright symbols throughout the text, including:

- ® on pages: 7, and 8

- TM on pages: 7, and 8.

5) Please upload all main figures as separate Figure files in .tif or .eps format. For more information about how to convert and format your figure files please see our guidelines:

6) We notice that your supplementary Figures, and Tables are included in the manuscript file. Please remove them and upload them with the file type 'Supporting Information'. Please ensure that each Supporting Information file has a legend listed in the manuscript after the references list.

7) Some material included in your submission may be copyrighted. According to PLOSu2019s copyright policy, authors who use figures or other material (e.g., graphics, clipart, maps) from another author or copyright holder must demonstrate or obtain permission to publish this material under the Creative Commons Attribution 4.0 International (CC BY 4.0) License used by PLOS journals. Please closely review the details of PLOSu2019s copyright requirements here: PLOS Licenses and Copyright. If you need to request permissions from a copyright holder, you may use PLOS's Copyright Content Permission form.

Potential Copyright Issues:

i) Figures 1, and S4A. Please confirm whether you drew the images / clip-art within the figure panels by hand. If you did not draw the images, please provide (a) a link to the source of the images or icons and their license / terms of use; or (b) written permission from the copyright holder to publish the images or icons under our CC BY 4.0 license. Alternatively, you may replace the images with open source alternatives. See these open source resources you may use to replace images / clip-art:

8) In the online submission form, you indicated that your data will be submitted to a repository upon acceptance. We strongly recommend all authors deposit their data before acceptance, as the process can be lengthy and hold up publication timelines. Please note that, though access restrictions are acceptable now, your entire minimal dataset will need to be made freely accessible if your manuscript is accepted for publication. This policy applies to all data except where public deposition would breach compliance with the protocol approved by your research ethics board. If you are unable to adhere to our open data policy, please kindly revise your statement to explain your reasoning and we will seek the editor's input on an exemption.

9) Please amend your detailed Financial Disclosure statement. This is published with the article. It must therefore be completed in full sentences and contain the exact wording you wish to be published.

10) Please send a completed 'Competing Interests' statement, including any COIs declared by your co-authors. If you have no competing interests to declare, please state "The authors have declared that no competing interests exist". Otherwise please declare all competing interests beginning with the statement "I have read the journal's policy and the authors of this manuscript have the following competing interests"

**Reviewers' Comments:**

Reviewer's Responses to Questions

**Part I - Summary**

Reviewer #1: Following the discovery of the Gp-Pat-1 gene, responsible for the emergence of virulence in G. pallida populations and the breakdown of potato resistance GpaVvrn, this manuscript examines the extent to which this genetic adaptation is conserved among a wider range of European populations.

The authors were able to identify an allele that is systematically selected by potato cultivars harbouring GpaVvrn and that this single SNP within the Gp-Pat-1 gene accurately reflects the reproduction obtained on GpaVvrn resistant potatoes. Based on this knowledge, the authors developed an AS qPCR test that was able to indicate the level of virulence of a population. Such a molecular tool will be of paramount importance both for PCN management in the field and for future research on the choice of additional leverage to combine with plant resistance or for a better understanding of the dynamics of selection and the highlighting of virulence costs during this adaptation process.

The manuscript is clear, well written, and the results presented are convincing.

Reviewer #2: This manuscript details the authors efforts towards identifying virulence factors and genetic makers in the agronomically significant pest, G. pallida. Using only nine different populations, they were able to confirm the presence of a previously identified marker. Then developed a qPCR assay specific for the virulence allele that was tested, blind, in four labs on 12 different populations. The authors were able to correlate allele frequency with population reproduction.

Reviewer #3: In the submitted manuscript, Schaveling et al. investigate how conserved the genetic basis of virulence against the GpaVvrn resistance is across West European Globodera pallida populations. G. pallida is an economically important pest of potato in many European regions. The most effective way to control this pest is by deploying host resistances, such as that of GpaVvrn from Solanum verneii. However, this can create a big challenge since the continuous use of resistant varieties can create strong selection pressure favouring virulence and leading to the emergence and increase in frequency of virulent alleles already present in the genetic pool. As a result, some populations may ultimately overcome resistance barriers. Previous studies have shown that all West European G. pallida populations originate from a specific Andean region and were introduced into Europe through potato imports in the 19th century. Based on this shared origin, the authors aim to determine how conserved this adaptation to resistance is across populations.

To address their research questions, the authors performed comprehensive and logically structured experiments. They first multiplied Dutch field populations for four generations on resistant potatoes containing GpaVvrn, generating sub-populations selected for increased virulence. Whole genome sequencing was performed on the whole set of populations at each generation, to track allele-frequency-related shifts that can be associated with selection evolution. To identify potential loci under selection, the authors focused on significant non-synonymous variants across genomic scaffolds. Any genomic regions enriched for such variants were put forward as predictive virulence loci. This approach led to the identification of a 300kb-long virulence-associated locus on the scaffold 28 that is consistent with the findings of two previous studies. Although the strategy of generating experimentally selected virulent populations has been applied before, the present study improves the genomic resolution of genomic regions associated with virulence in G. pallida as well as strengthens the evidence for their role in adaptation.

After confirming the virulence-associated locus, the authors assessed the alternative allele frequencies (AAF) of the gene Gp-pat-1 using genomic data from a diverse range of West European populations (from the UK and France) that had previously shown to overcome GpaVvrn resistance, alongside with the generated Dutch sub-populations. The gene Gp-pat-1, recently identified by the same research group, resides with the identified locus and contributes to virulence against the resistance GpaVvrn. The authors found consistently elevated AAF values in 3 variants of this gene across the majority of the analysed West European populations. These results strongly recommend that virulence against this resistance source has a shared basis, consistent with the common geographic origin of these populations. This finding is novel and highly significant, as it gives the possibility to the research and breeding industry community to use these variants as molecular markers for virulence in the future.

Indeed, to functionally validate this association, the authors conducted two standardised PCN phenotyping tests using eight Dutch populations on 28 GpaVvrn-containing potato varieties. The phenotypic data showed that the AAF of the SNP T173N (one of the four SNPs of the virulence locus), correlated with increased G. pallida reproduction rate. Therefore, the authors propose that this SNP is a reliable marker for predicting virulence. They subsequently developed an allele-specific qPCR assay to estimate AAF values and applied to field populations to predict virulence status. Using this assay, they screened Dutch and German field populations and validated the results by cross-referencing qPCR-derived AAF estimates with AAF values from sequencing analyses, revealing strong correlations. This assay was repeated by different labs and using different DNA extraction protocols to validate its robustness.

The ability of G. pallida populations to overcome certain resistance sources is a major challenge for the European potato industry, making this study both timely and highly relevant. Overall, the experimental design was rigorous, the methodologies were well-thought, and the study demonstrates a high level of technical and analytical quality. The authors also used clear and logical rationale. By applying these genomic approaches, the authors offer new insights regarding the understanding of the genetic basis of virulence in this genetically complex plant parasitic nematode. Importantly, their findings also extend our knowledge of the mechanisms underlying resistance breakdown, offering findings of high novelty and potential significance for fundamental and applied research. Especially, the development of a molecular assay for predicting virulence represents a strong and innovating aspect of their work, with a big potential to be integrated as a high-throughput diagnostic tool to support resistance management and decision-making in agriculture.

**Part II – Major Issues: Key Experiments Required for Acceptance**

Reviewer #1: (No Response)

Reviewer #2: None

Reviewer #3: No major issues have bee identified in this manuscript.

**Part III – Minor Issues: Editorial and Data Presentation Modifications**

Reviewer #1: I fully support the publication of this manuscript after the implementation of a few minor comments below:

- Lines 57-58: I am not sure that a blocking resistance necessarily acts after sex determination. This sentence should be reworded, and the authors should also take into consideration certain blocking resistances that also act before sex determination (such as male bias resistance). See, for example, the resistance obtained with the potato carrying the two S. sparsipilum QTLs for resistance (Caromel et al. 2005, MPMI) or even, in the case of RKN, the Me3 gene in pepper (Bleve-Zacheo et al., 1998, Plant Sci), which is a blocking resistance acting during nematode migration within the roots and before any induction of the feeding site.

- Lines 158-159: please indicate the number of replicates used in the resistance tests so that readers who are not familiar with the EPPO standard protocol can have this information.

- Suplementary Figure 1 and suppl Figure 2 : please add in the legends of these two figures the names of the populations corresponding to A, B, C, D, …

- I would have preferred to see some of the information presented in Supplementary Text 1 included in the main text. In particular, the information and results related to the influence of AAF and the number of cysts to be used to obtain a reliable result. These data are of great importance as they highlight certain limitations of the AS-PCR tool. Another limitation that is not addressed in the current manuscript and would have been interesting to examine is the sensitivity of the tool. Did the authors consider using the tool developed at the level of a single cyst or J2? As Gp-pat-1 is a single-locus gene, this means that genotyping using such a tool should allow the homozygous vir or avir or heterozygous status of individuals to be determined.

- Still regarding this issue of tool sensitivity, I noticed that the authors used primers with LNA. Somewhat unexpectedly (LNA is expected to bring more specificity and sensitivity), the standard allele-specific primers gave a better indication of AAF than the LNA primers. I would have liked to see a discussion of this result. Is it related to the sensitivity of the tool? Did the authors try testing the primers with LNA on small amounts of DNA?

Reviewer #2: The authors use "across western Europe", but may need more sampling to support this claim. As it stands, it is fine, but I urge caution on such broad claims as this region includes more than Netherlands, England and France.

Are the nematodes Dutch as well? Should it be G. pallida populations from the Netherlands?

Reviewer #3: Some minor issues have been identified that require some modifications or further explanations. Please see specific comments per section below:

Introduction:

Introduction is well-structured with good background introduced.

Methods:

L94: Reference to the tidyverse package needs to be added.

L106: While the populations were grown for four generations on the resistant Seresta, the control that was used in the study was grown only for one generation on the susceptible Desiree. It is understood that no drift is expected in the susceptible variety, but would not be a good idea to use the same multiplication approach for both selected and control populations? Could the authors please support their decision?

L160 (“At the end of the experiment…”): Can the authors please specify how long (days) post inoculation this means?

L247-248 (“however,…can reach up to 100%”): Can the authors please detail why the virulence level can be overestimated compared to the standardised pot trials? Citation might be the wrong one, as it was not mentioned by Price et al. (2024).

Results:

The sections “The alternative allele frequency of a single SNP correlates with the reproduction rate on GpaVvrn” and “Allele-specific qPCR confirms GpaVvrn-mediated selection of T173N” are too long and a bit confusing to the reader. Although the reasoning behind is good, it would be helpful to break them into smaller sentences to help the reader focus on better.

This is especially a case for the second paragraph (L337 – L369) that is a bit difficult to follow. It also feels that as it is written now, it would be better suited to Discussion rather than Results. I would recommend adding some findings (in quantitative terms, e.g. percentages) directly to the text (e.g. how different are the AAF between unselected-selected populations found from qPCR, etc.) on top of referring to the Figures (and Supplementary). This will significantly keep the interest and the focus of the reader.

L371 – L372: This will need a further explanation in the text. Referring to supplementary results only can be confusing to follow.

General:

Across the whole manuscript, the number of tested populations change depending on the experiment. For example, 22 populations were tested for the robustness of the qPCR assay whereas only 8 in the phenotyping tests. Can the authors please add some clarifications why this was chosen (maybe in the Discussion or Methods)? This will enhance the clarity of the manuscript even further. In addition to that, a table showing which populations were used in which experiment could also be added as Supplementary.

Supplementaries:

L513: In the section “Statistical analysis on the reliability of the AS-qPCR assay”, the authors assume that the populations follow the Hardy-Weinberg (HW) equilibrium in order to calculate the standard error used for comparing the qPCR-derived AAF values with those from sequencing analyses. However, one of the assumptions of HW equilibrium is the absence of natural selection, meaning that no genotype/individual is favoured. In this study, the generated populations were subjected to strong selection pressure for a number of generations, and virulent nematodes were probably favoured during the evolutionary process, which is contradicting to the HW equilibrium assumptions. Could the authors please justify/support further the use of this assumption in this context in order to clarify their approach?

PLOS authors have the option to publish the peer review history of their article (what does this mean?). If published, this will include your full peer review and any attached files.

Reviewer #1: No

Reviewer #2: No

Reviewer #3: No

**Figure resubmission:**

After uploading your figures to PLOS’s NAAS tool - https://ngplosjournals.pagemajik.ai/artanalysis, NAAS will process the files provided and display the results in the "Uploaded Files" section of the page as the processing is complete. If the uploaded figures meet our requirements (or NAAS is able to fix the files to meet our requirements), the figure will be marked as "fixed" above. If NAAS is unable to fix the files, a red "failed" label will appear above. When NAAS has confirmed that the figure files meet our requirements, please download the file via the download option, and include these NAAS processed figure files when submitting your revised manuscript
---

## [Editor Report · Decision Letter 1]

25 Apr 2026

Dear Mr. Sterken,

We are pleased to inform you that your manuscript '*Globodera pallida* virulence on major potato resistance has a common genetic basis across Western Europe' has been provisionally accepted for 'publication in PLOS Pathogens.

Best regards,

Adler R. Dillman, Ph.D.

Academic Editor

PLOS Pathogens

Shou-Wei Ding

Section Editor

PLOS Pathogens

Sumita Bhaduri-McIntosh

Editor-in-Chief

PLOS Pathogens

orcid.org/0000-0003-2946-9497

Michael Malim

Editor-in-Chief

PLOS Pathogens

orcid.org/0000-0002-7699-2064

The authors have done a commendable job addressing the minor revisions requested by the reviewers.
---

## [Editor Report · Acceptance letter]

Dear Mr. Sterken,

We are delighted to inform you that your manuscript, "*Globodera pallida* virulence on major potato resistance has a common genetic basis across Western Europe," has been formally accepted for publication in PLOS Pathogens.

Best regards,

Sumita Bhaduri-McIntosh

Editor-in-Chief

PLOS Pathogens

orcid.org/0000-0003-2946-9497

Michael Malim

Editor-in-Chief

PLOS Pathogens

orcid.org/0000-0002-7699-2064